# Exploring barriers to parent-adolescent sexual-risk communication among adolescents in Port Harcourt Nigeria: Adolescents' and parents' perspective

Chidinma Joycelyn Okpalaku[1, *], Chigozirim Ogubuike[1, 2]

1 School of Public Health, University of Port-Harcourt, Port-Harcourt, Rivers State, Nigeria, 2 Department of Community Medicine, University of Port-Harcourt Teaching Hospital, Port-Harcourt, Rivers State, Nigeria

* chidinma.okpalaku@gmail.com

## Abstract

Adolescent risky sexual behaviour is a public health problem with its deleterious outcomes. Parents are the most influential source of sexuality education to adolescent, yet adolescents' lack sexuality educations. The study explored barriers in parent-adolescent sexual-risk communication from both perspectives in Port-Harcourt LGA, Rivers State. A cross-sectional study design using explanatory sequential mixed methods approach was implemented. Three hundred and twenty nine in-school adolescents participated in the quantitative study and recruited using multi-stage sampling technique while 9 parents of adolescents and 16 adolescents participated in the qualitative study. A self-administered questionnaire was used to elicit information from the adolescents while an FGD and IDI guide was used to elicit information from in-school adolescents and parents respectively. The quantitative data was analysed using descriptive statistics and chi-square while the qualitative data was subjected to thematic analysis. The mean age of the adolescents was 16.0 ± 1.1 years and 55% were males. 21% of the parents had never discussed sex with their adolescents. The barriers identified from the adolescents' perspective were parental factors (parents being too busy, judgmental, low knowledge), individual factors (discomfort to initiate communication, lack of trust), religious and cultural factors. The barriers from the parents' perspective were shame to initiate communication, fear of outcome, feeling children are too young and lack of accurate information. The barriers to parent-adolescent communication featured interplay of parental, individual, cultural and religious factors. Parents should be trained to initiate timely and accurate sexuality education to the adolescent to curb adolescent risky sexual exploitations.

## Background

Adolescence, which occurs between the ages of 10 and 19 years, is a stage between childhood and adulthood marked by profound psychosocial, psychological, and social changes [1,2]. Adolescents constitute of 1.8 billion or one-fourth of the world's population, therefore the care for this group is accompanied with socioeconomic progress and societal advancement that are

**Data availability statement:** The data supporting the findings of this study are not openly available due to ethical reasons, specifically the inclusion of underage human participants. However, the data can be made available from the corresponding author upon reasonable request. For inquiries related to data access, please contact the following individual, who is not involved in the research and is authorized to handle data requests: Daniel Okon (Email: daniel.okon@uniport.edu.ng; Phone Number: +234(0)7067281841). Additionally, the data will be stored securely, and procedures for long-term storage and availability will be managed in accordance with institutional guidelines.

**Funding:** The authors received no specific funding for this work.

**Competing interests:** The authors have declared that no competing interests exist.

being confronted [3]. They go through anatomical and psychological changes that have a profound impact on their mental wellness, which is manifested as high levels of sexual curiosity [4]. This sexual curiosity could result in risk-taking sexual behaviour if there is no behaviour counselling during the adolescent years [5]. Risky sexual behaviours are behaviours that put a person at risk for STIs like HIV and unintended pregnancies. These actions include having several sexual partners, transactional sex, forced or coerced sex, and unprotected sexual contact [6]. Adolescents who are sexually active are more likely to participate in risky behaviours that could increase the chance of unintended pregnancy and abortion, and increase the risk of sexually transmitted diseases [7,8].

Researches have shown that open sexuality communication between parents and children can significantly lower the sexual risk [9,10], protect against non-risky sexual behaviours, such as a delayed sexual debut, engaging in safe sex, and abstinence especially in females [11] and lower the morbidity and mortality associated with risky sexual behaviours [12]. In addition, parent-child sexual communications have been shown to increase awareness and solve problems associated with sexual and reproductive health [13,14]. Adolescents' knowledge about sex education is primarily from classmates and schoolmates, who often provide inaccurate information [15]. Sexual behavior is a sensitive issue in churches, schools, and families, with each party pointing fingers at each other when problems arise [16].

Most parents often struggle with discussing sexual and reproductive health issues with their children, often focusing on "safe" topics or providing cautionary statements [17]. This lack of communication can lead to serious social and health consequences, such as parent-child conflict, school dropouts, unwanted pregnancy, unsafe abortion, and STIs like HIV and AIDS. Additionally, parents often have a negative aversion to sex education due to cultural and religious influences from their fore parents [18]. A study conducted in Ghana revealed that parental and societal distaste to sex education hasn't changed over time which has resulted into an overlooking of adolescents as they receive virtually no or relatively little reproductive health care and education [19]. In many Nigerian households, adolescents lack access to informative and undiluted sexual communication with their parents and have adopted indirect communication patterns [20]. Nearly half of the adolescents in Port-Harcourt had sexual experience with multiple sexual partners [21], hence the study locale. However, there is paucity of data on the barriers to adolescent-parents sexual-risk communication in Port-Harcourt, Nigeria.

Therefore, this study aimed to understand parent-adolescent sexual risk communication from both parents and adolescents perspectives to identify barriers and provide recommendations for easier communication. The research emphasized the significant influence of parents on adolescents' sexual education, focusing on the proportion of adolescents receiving sexuality education from their parents and identifying barriers to effective communication. The outcome of this study would help in policy and programmatic directions curb risky sexual exploitations in adolescents.

## Materials and methods

### Study area

Port-Harcourt city is the capital of Rivers State in Nigeria otherwise known as Port Harcourt Local Government Area (PHALGA). It is one of the 23 local government areas created for the state. It is located in the southern part of the country and is one of the states in the Niger Delta region. The Local Government Area is bounded by Ikwerre and Etche Local Government Areas at the North, Asari-Toru and Okrika Local Government areas at the South, Emohua LGA at the West and Eleme and Oyigbo LGAs at the East in Rivers State [22]. In 2021, the population of

Port Harcourt was estimated to be 3,171,076 [23]. Port Harcourt is the nerve centre of the Nigerian Oil industry and includes other industries too. It is also the nation's second largest sea port with the Onne Port Complex in close proximity. Marine agriculture is the main occupation of the people of Rivers state with farming and trading being the two major occupations of Port Harcourt residents. It is the fifth most populous city in Nigeria after Lagos, Kano, Ibadan, and Benin. The state has a 154 non-tertiary public, private and government schools which means a whole lot of adolescents enrolled into these schools which covers the study population for this study. These adolescents constitute a large part of the State's population.

## Study design

A cross-sectional study design using the explanatory sequential mixed method approach was utilized for this research. It is a two-phase mixed methods design that starts with the collection and analysis of quantitative data followed by the subsequent collection and analysis of qualitative data. This method was used to enable the researcher explore the research problem quantitatively and then explain it qualitatively.

## Study participants

The study participants involved in-school adolescents and parents of adolescents. This study included in-school male and female adolescents between the ages of 15–19 years from junior secondary school (JSS3) and senior secondary school (SS1, SS2 and SS3) in selected schools in Port Harcourt LGA. Parents recruited were those that have adolescents. The adolescents were involved in both quantitative and qualitative data collection while the parents were involved in only qualitative data collection. The adolescents were identified face to face in their school and recruited for the survey while the parents were approached during school PTA forum and recruited for the study. All the interviews were conducted face to face and audio recorded.

## Research team and flexibility

The interviews were conducted by the researchers who are females with background training and experience in qualitative research. One of the researchers already had Masters in public health and the other was currently a postgraduate student in public health. One of the researchers conducted the interviews while the other served as a note taker. The researchers formerly introduced themselves to establish a rapport before data collection commenced. The study participants were informed about study goal and objectives, benefits and significance of the study and they were free to ask questions.

## Sample size determination

**Quantitative.** The sample size for the quantitative aspect was determined using Cochran's formula:

$$n = \frac{z^2 pq}{d^2}$$

where,

n = the desired sample size.

z = the standard normal deviate, usually set at 1.96 which corresponds to 95% confidence level.

p = the proportion of adolescents who communicated with their parents on SRH issues was 36.9 or 37% [15].

q = 1−p = 1−0.37 = 0.63

d = degree of accuracy desired, usually set at 0.05.

n = $1.96^2 \times 0.37 (1-0.37)/0.05^2$ = 358. 1. Assuming a non- respondent rate of 10%, the minimum sample size was estimated at 394.

**Qualitative.** For the qualitative aspect, 9 parents of adolescents and 16 adolescents were purposively recruited to participate in an in-depth and focus group discussion respectively. The focus group comprised of two groups (a male and female group) with 8 participants each. Interviews were stopped when saturation was achieved. Two parents refused to participate in the study because of busy work schedule.

## Sampling method

**Quantitative.** For the quantitative aspect a multistage sampling method was used in this study. The first stage involved random selection of schools. Permission and the list of schools in Port Harcourt LGA was obtained from the Rivers State Ministry of Education and used as the sampling frame. The schools were first stratified into private and public schools and simple balloting was used to select two private and two public schools. The second stage is the selection of students according to classes and gender. The students were stratified into 3 groups based on their classes, i.e., JSS3, SSS1, SSS2, and SSS3 and then re-stratified based on their sexes (Males and Females). The third stage involved the proportionate allocation. The sub-sample for each stratum was divided proportionately into the number of arms/forms in each class. The sample size was proportionately divided into each of the strata based on their population. About 100 students were recruited from each school, hence a total of 400 participants were recruited.

**Qualitative.** In the qualitative aspect the respondents were recruited purposively, the adolescents participated in a focus group discussion (FGD) [Data A and Data B in S1 Text] while the parents participated in an in-depth interview (IDI) [Data A to Data I in S2 Text]. FGD was conducted to explore group dynamics and gather diverse perspectives of adolescents on the subject matter. In-depth interviews were conducted for in-depth exploration, to get personal experiences and opinions on parents' perceptive of barriers.

## Data collection

The recruitment of participants for the pre-test commenced on the 5th–7th May 2021, furthermore recruitment of participants for the study was between 6th–24th July 2021. Parents were recruited 6th–11th August 2021. Self-administered questionnaires (S1 Appendix) were given to the adolescents which they completed themselves. Data was gotten from the parents via in-depth interview (Text A in S1 File) and was audio recorded. The consent of the participants was sought before the research questionnaire was given. Participants were asked to sign on a well-written consent form enclosed on the questionnaire after the study had been explained to them so as to show their voluntary participation. For adolescents below 18 years, it was ensured that their guardians in form of the school principals were present, signed the consent forms and the adolescents also assented before questionnaires were administered. The questionnaires were collected after being completed and the participants were appreciated for their contribution and time. There were no repeat interviews. The focus group discussions (Text B in S1 File) lasted for about 60 minutes while the in-depth interview lasted for about 30 minutes.

## Study instrument and data analysis

A pre-tested structured self-administered questionnaire was used to elicit information on the socio-demographic characteristics of the respondents, source of information and proportion

of adolescents that received sex education. The researcher administered the questions to the adolescents and retrieved them after completion. An in-depth interview guide and focus discussion guide was used to elicit qualitative information on the barriers and facilitators to parent-adolescent sexual-risk communication. The questionnaire was adapted from earlier studies on parent-adolescent communication about sexuality [17,24,25] while some other questions were introduced by the researchers and given to two experts to check for face validity. SPSS statistical software version 25 was used for the quantitative analysis. Mean and standard deviation were used for descriptive statistics. Atlast ti software was used for the analysis of the qualitative data. The interviews were audio recorded, transcribed verbatim, carefully read and edited by the researchers to ensure original meanings were not lost to translation. The transcripts (Data A to Data I in S2 Text, Data A and Data B in S1 Text) were read and re-read to develop a preliminary understanding of the data, coding was done and a coding framework (File A and File B in S1 Checklist) was developed, codes were categorized, and organized and themes identified and refined. Thematic analysis was done by the researchers.

## Ethical considerations

Ethical clearance was obtained from the Research and Ethics Committee of the University of Port Harcourt (UPH/CEREMAD/REC/MM74/038) and permission obtained from the school authority of chosen schools before commencement of the study. A verbal consent was also obtained from the parents/guardians for adolescents participating in the quantitative study during PTA forum before commencing the study. The assents of the adolescents' were sought before the research questionnaires were distributed. In the qualitative aspect, for adolescents below 18 years, it was ensured that their guardians, in the form of the school principals, were present, signed the consent forms, while adolescents 18 years and above signed the consent forms. Parents involved in IDI were asked to sign a well-written consent form after the study had been explained to them, so as to show their voluntary participation. Information of the study was made available on each consent form. Confidentiality was guaranteed as personal details of respondents were not included in the data collection instrument. Serial numbers were used and as a result, collected data wasn't linked to any of the respondents.

## Results

### Socio-demographic characteristics

**Socio-demographic characteristics of the adolescents.** The socio-demographics of adolescents recruited into the study is shown in Tables 1 and 2. A total of 329 out of 400 students completed the questionnaire giving a response rate of 83.5%. The result revealed that 236 (71.7%) of the adolescents were between the ages of 15–16 years with a mean age of 16.0 ± 1.1 years, 190 (57.8%) were males, 165 (50.2%) were in SS2 and 316 (96.0%) of the respondents were Christians. Most of the adolescents 223(67.8%) belonged to a monogamous family and 202(61.4%) lived with their biological parents. About 196(59.6%) of the adolescents had a family size of 4–6 persons, 129(39.2%) of the adolescents' fathers had only vocational training, 102(31.0%) of the adolescents mothers had no formal education, 145(44.1%) of the adolescents' fathers were self-employed and 204(62.0%) of the adolescents' mothers were self-employed. For the qualitative data, 16 adolescents participated in an FGD of two groups (8 males and 8 females).

**Socio-demographic characteristics of parents.** Nine parents of adolescents participated in an IDI session comprising of 5 males and 4 females among which 3 were teachers, 3 traders and 3 civil servants. Six of them had attained a tertiary education and three had attained a secondary education.

**Table 1. Socio-demographic characteristics of adolescents in Port Harcourt, Rivers State.**

| Variable | Frequency n = 329 | Percentage |
|---|---|---|
| **Age group** | | |
| 15–16 years | 236 | 71.7 |
| 17–19 years | 93 | 28.3 |
| *Mean age* | *16.0 ± 1.1* | |
| **Sex** | | |
| Male | 190 | 57.8 |
| Female | 139 | 42.2 |
| **Class** | | |
| SS1 | 136 | 41.3 |
| SS2 | 165 | 50.2 |
| SS3 | 28 | 8.5 |
| **Religion** | | |
| Christianity | 316 | 96.0 |
| Islam | 10 | 3.0 |
| Traditionalist | 3 | 0.9 |
| **Family type** | | |
| Monogamy | 223 | 67.8 |
| Polygamy | 38 | 11.6 |
| Single parent | 55 | 16.7 |
| Step parent | 13 | 4.0 |
| **Family structure** | | |
| Two Biological parents | 202 | 61.4 |
| Biological father only | 15 | 4.6 |
| Biological Mother only | 34 | 10.3 |
| Biological father and step mother | 9 | 2.7 |
| Biological mother and step father | 5 | 1.5 |
| With guardian | 52 | 15.8 |
| Other family member (grandparents) | 12 | 3.6 |

## Proportion that received sexual education from parents

Table 3 shows the proportion of adolescents that received sexual education from their parents.

It was shown that 69(21.0%) of the parents had never discussed sex education with their children, 107(32.5%) of adolescent reported that they would be more comfortable discussing sexual related issues with their mother. A majority of them 190 (57.8%) received sexual education from school.

## Perceived barriers to parent-adolescent sexual-risk communication: Adolescents' perspective

The perceived barriers to effective sexual-risk communication outlined by the adolescents were grouped into three broad categories; parental, individual and religious/cultural factors. Parental factors involved unavailability of parents/parents being too busy, parents are too judgmental, parental poor/inaccurate knowledge on sex education, parental belief that discussion will initiate sex. Individual factors were lack of trust, fear and discomfort of sexual communication.

**Table 2. Social demographic characteristic of adolescents in Port Harcourt, Rivers State.**

| Variable | Frequency n = 329 | Percentage |
|---|---|---|
| **Family size** | | |
| Less than three | 42 | 12.8 |
| Four to Six | 196 | 59.6 |
| Seven to nine | 62 | 18.8 |
| 10 and above | 29 | 8.8 |
| **Father's education** | | |
| No formal education | 129 | 39.2 |
| Vocational training | 118 | 35.9 |
| Primary | 29 | 8.8 |
| Secondary | 37 | 11.2 |
| Tertiary | 16 | 4.9 |
| **Mother's education** | | |
| No formal education | 102 | 31.0 |
| Vocational training | 141 | 42.9 |
| Primary | 27 | 8.2 |
| Secondary | 41 | 12.5 |
| Tertiary | 18 | 5.5 |
| **Father's occupation** | | |
| Employed by the government | 79 | 24.0 |
| Private employment | 91 | 27.7 |
| Self-employment | 145 | 44.1 |
| Unemployed | 14 | 4.3 |
| **Mother's occupation** | | |
| Employed by the government | 55 | 16.7 |
| Private employment | 52 | 15.8 |
| Self-employment | 204 | 62.0 |
| Unemployed | 18 | 5.5 |

**A. Parental factors.** *Unavailability of parents*: Most of the adolescents reported that their parents were too busy with work and rarely had time to communicate with them. Even when parents were present, they were unavailable to have communications with their wards. One of the male adolescent reported:

> *"My dad is hardly ever home and even when he is at home; he says he's too tired to have any conversations with me. He doesn't have our time. All of my sexual knowledge is either from school, my friends or from the internet".* **Male Adolescent (16–20 years)**

A female adolescent narrated serious obstacles in discussing with her mother and not able to have sexual communications with her father due to gender barriers. She mentioned receiving sexual education from the school by their teachers, peers and the internet.

> *"My mom is always travelling for business so, discussing sex with her is almost impossible. I can't discuss with my dad because he is of the opposite sex. Most of the knowledge I have on sex is what I was taught in school by my teachers, some from my friends too".* **Female Adolescent (11–15 years)**

Table 3. **Proportion of adolescents in Port Harcourt, Rivers State that received sexual education from parents/guardians.**

| Variable | Frequency n = 329 | Percentage |
|---|---|---|
| **How often parents/guardian discuss sex education** | | |
| Never | 69 | 21.0 |
| Rarely | 91 | 27.7 |
| Sometimes | 123 | 37.4 |
| Often | 20 | 6.1 |
| Always | 26 | 7.9 |
| **Comfortable discussing sexual related issue with** | | |
| Father | 35 | 10.6 |
| Mother | 107 | 32.5 |
| Siblings | 31 | 9.4 |
| Teachers | 17 | 5.2 |
| Peers of the same sex | 50 | 15.2 |
| Friends | 89 | 27.1 |
| **Source of sex education** | | |
| Home | 105 | 31.9 |
| School | 190 | 57.8 |
| Internet/social media | 25 | 7.6 |
| Church | 9 | 2.7 |

*Parents too judgmental*: Both the male and female adolescents in their different discussion groups agreed that their parents were too judgemental, read meaning into little things and condemn them when they asked questions on sexual and reproductive health related matters. They resorted never to have such discussions again with their parents. One of the female adolescent reported that:

*"I don't discuss sexual matters with my mom because she will judge and condemn me, then assume that I want to start having sex so, I don't bother meeting her to start up such discussions. She judges a lot and condemns me". Female Adolescent (16–20 years)*

A male adolescent averred that he wouldn't have any other sexual communications with his parents because of the way his parents handled the question he asked on condom efficacy in prevention of pregnancy:

*"My parents are the kind of people that read meaning into everything; even mere questions. My asking questions about these issues will mean that I have already started having sex or I want to have sex. There was this time I asked my dad if condoms were 100% effective in preventing pregnancy....My dad called my mom immediately and they started asking me all sorts of really embarrassing questions. Since then, I never asked them such questions again". Male Adolescent (16–20 years)*

*Parental belief that discussion will initiate sexual intercourse*: Parental assumptions that sexual education leads to initiation to sexual intercourse was reported by the adolescents as barriers to sexual-risk communications. One of the adolescent avowed that their parents believed sexual discussions would expose them to sexual exploitation. He stated:

*"Whatever information I got first about sex was from the school. My parents have never discussed sexual issues with me because they probably feel that I would want to venture into sex". Male Adolescent (11–15 years)*

In affirmation, another male adolescent narrated the ugly experience melted on him by his parents when he made enquiries on sex and condom use. He narrated:

*"The first time I asked my dad about sex and condoms, he was very angry and asked me if I had already started having sex or want to have sex. He almost beat me up because of that one simple question. I was only 14 at the time and as a result I vowed never to discuss sexual issues with my dad and he doesn't discuss with me either. I learn from peers and the internet".* **Male Adolescent (16–20 years)**

**Parents lack of knowledge on SRH**: Adolescents' reported that their parents were not well equipped to give updated information on sexual and reproductive health issues hence they shy away from such discussions with them. They also narrated meeting their peers and the internet for information. A male adolescent reported:

*"I don't discuss sexual issues with my parents especially my mom because they are both 'old school'. Whatever information they have will be archaic. It's either I discuss with peers/friends or use the internet".* **Male Adolescent (16–20 years)**

One of the female adolescents affirmed that the sexuality knowledge of their parents' were archaic/out-dated but the school gives more updated information. She avowed:

*"I don't think my parents are well equipped to give me adequate knowledge on SRH issues. They are archaic. They do not just have enough knowledge. At least I receive up-to-date information in school from my teachers, peers and sometimes from the internet".* **Female Adolescent (11–15 years)**

**B. Individual factors. Lack of trust**: Lack of trusting their parents, lack of trust by their parents and lack of trust parents had on their friends was reported by the adolescents as hindrances to sexual risk communication by their parents. One of the female adolescent elaborated:

*"I don't trust my parents and my parents don't trust my friends because they feel they will influence me negatively and as a result they do not trust me either. I have been warned against having a boyfriend or having sex. If I dare broach the subject of sex to them, then I'm looking for trouble".* **Female Adolescent (11–15years)**

**Feeling of discomfort with sexual discussion**: Sexual risk communication discomfort was reported by the adolescents as barriers to sexual education. As a result of this discomfort they sourced sexual and reproductive information from their friends. One of the female adolescents reported that they felt uncomfortable having sexual discussion with their parents.

*"Whenever I ask my Mom about sexual issues, she gets so uncomfortable with the discussion and it makes me shy. After a while I stopped discussing with her but I discuss with my friends in school".* **Female Adolescent (16–20 years)**

A male adolescent asserted same, that their parents felt very uncomfortable too with such discussions.

*"My parents rarely discuss sexual issues with me because they are always so uncomfortable with the discussion which also makes me uncomfortable too. I think we all noticed this and*

*stopped its discussion completely. They don't discuss sexual matters with me anymore".* **Male Adolescent (16–20 years)**

***Fear to initiate sexual discussion:*** Some of adolescents reported being selective on sexual and reproductive health issues to discuss with their parents as a result of fear associated with sexual discussions. One of the female adolescents averred she only discussed selective topics with her mother because they were of same gender:

*"I discuss more of SRH issues like physical development, menstruation with my mom because we are of the same gender. I dare not mention sex or any related topic to her because she will beat me. She will think that I have a boyfriend and want to experiment with him".* **Female Adolescent (16–20 years)**

A male adolescent reported being too scared to initiate discussion with their parents because of negative assumptions it may generate. He said:

*"I would love to discuss sexual issues with my parents but I am too scared of what they will think or feel about me".* **Male Adolescent (16–20 years).**

**C. Religious/cultural factors.** Adolescents reported that their parent considers sexual education as a taboo as a result of their religious and cultural affiliations. One of the female adolescent narrated that sexual communications were forbidden at home due to religiosity:

*"....my parents are just too religious. Such discussion is a forbidden at home. If I even ask them by mistake, they will skin me alive because they will assume that I want to start experimenting. I'm not even allowed to have male friends around me".* **(Female Adolescent, 16–20 years old)**

A male adolescent affirmed that parents do not discuss sex with them because it is against their religious affiliations.

*"My parents do not discuss sexual issues with me or my siblings because it is against our religion. Our Church forbids such discussions in the home. Whatever I know about sexual issues, I got from my friends in school".* **Male Adolescent (16–20 years)**

Cultural factors were highlighted as a hindrance to sexuality education. A female adolescent avowed that:

*"....whenever I ask my mom about issues relating to sex, she always tells me it is against our culture to discuss such issues with children".* **Female Adolescent (11–15 years)**

## Perceived barriers to parent-adolescent sexual-risk communication: Parents' perspective

Parents had varied opinions on the necessity of discussing sexual and reproductive health issues with adolescents. Some believe that children may make mistakes without education, while others believe it is unnecessary. They believe discussions will provide children with ideas on sex and initiate early sexual experimentation, as they tend to listen more to their parents. One of the parent narrated negative consequences of non-sexuality communication leading to unintended pregnancy and school dropout:

*"I think parents should discuss with their children. My neighbour's daughter got pregnant at 16 and that brought shame to the family. She had to quit going to school at a point as a result of this. Maybe, if her parents had discussed these things with her, she won't have made such a mistake. As a result of this, I had to start discussing sex with my children. Children should be armed with the knowledge to make wise decisions regarding their sexual life".* **Male Parent of Adolescent, Civil Servant**

Parents narrated their perceived barriers to sexuality communication with the adolescents. These barriers were grouped into three factors namely; Individual factors (shame initiating sexuality discussion and gender barriers, barely enough time for discussion, level of education of parents), adolescent factors (fear that children will want to experiment sexually, feeling that children are too young to be educated on SRH issues), and tradition/religious beliefs.

**A. Individual factors.** *Shame initiating sexuality discussion and gender barriers*: Parents revealed that they sometimes experience a feeling of shame while discussing SRH issues with their children. A female parent reported shame and adhering to intergenerational patterns of non-sexual communications:

*"I have never discussed things like puberty with its associated changes in males and females with my children because I feel so shamed discussing these issues with them it is an abomination in my culture too. My mom never discussed with me so, I never do with my children".* **Female Parent of Adolescent, Petty Trader**

Gender barriers was reported as factors that hinder parents from effectively discussing sexual and reproductive health issues with their adolescents as revealed in the study. Parents fail to discuss with their children of the opposite sex on sexuality education such as condom use, puberty and physical development. A female parent averred that:

*"I have 2 sons who I can't communicate on sex, condom use or things like 'wet dreams' because it is shameful for me. I leave this discussion for my husband to do".* **Female Parent of Adolescent, Teacher**

A male parent claimed that sex education is purely the role of the mother as he feels awkward engaging in such conversations:

*".....I really feel is awkward discussing with my daughters especially on puberty and menstruation. It is the job of their mother and their teachers".* **Male Parent of Adolescent, Civil Servant**

*Barely enough time for discussion*: Another barrier that was highlighted by parents was the issue of time. Some of the parents do not discuss with their children because they are just too busy with work and so the discussion is usually left to the other spouse. A male parent gave narrated not having enough time for sexuality education and abandoning the sex education for their spouse who may barely have the time to initiate such discussion:

*".....my kind of job doesn't give me enough time to discuss with my children. I'm usually really tired when I get home and I work most weekends. I only make out time to discuss really important stuff with them".* **Male Parent of Adolescent, Civil servant**

*Level of education of parents*: Parents who had a higher level of education or work in a corporate establishment discuss more about SRH especially sexual issues than others who had a lower level of education. A male parent with secondary school education reported that:

*"I have never discussed sex education with my children because I feel they shouldn't know about these kinds of things. What they don't know will not 'spoil' them".* **Male Parent of Adolescent, Trader**

**B.  Adolescent factors.  *Fear that children will want to experiment sexually*:** Parents feared that discussion about pregnancy, how to practice safe sex, condom use and sexually transmitted infections will push or encourage their children into engaging in sexual activities. A parent perceived that sex education with adolescents will enhance their sexual exploitations:

*"Discussing issues like condom use, sex, STIs/STDs will make my sons feel that they can venture into sex even before they come of age. This is why I don't discuss these with them".* **Female Parent of Adolescent, Teacher**

Another parent affirmed same:

*"Discussing things like pregnancy and prevention, relationships etc. will make my daughter feel that it's okay to have sex in her relationships with the opposite sex probably. Children of this age just want to experiment and besides, she is also too young to understand".* **Female Parent of Adolescent, Trader**

***Feeling that children are too young to be educated on SRH issues*:** Most parents do not discuss SRH issues with their children because they feel that their children are too young to be taught SRH issues like sex, pregnancy, condom use, STIs/STDs. A female parent expressed that early adolescents are too young to receive sexual education from their parents:

*"My daughter is too young for this sort of discussion and I don't really feel comfortable discussing these kinds of things with her. How can I start educating my 11 year old daughter about pregnancy, sex or even STIs? I can't discuss these with her at such a young age".* **Female Parent of Adolescent, Petty Trader**

**C.  Tradition/religion.**  Tradition and religion were also seen as barriers to effective parent-adolescent sexual risk communication. A male parent inferred that they do not discuss sexual issues with their children because it is against their tradition.

*"For me it's against my tradition to discuss these things with my children especially my son who is too young. He's only 15. I feel he should know stuff like these when he's officially an adult".* **Male Parent of Adolescent, Trader**

Another parent extrapolated that she only discussed menstruation and physical development with her daughter due to religious restrictions.

*"I have never discussed sexual issues with my children because the church frowns at it. I only discuss things like menstruation with my daughter because she needs to know about it and physical development for all my children because they all need to know".* **Female Parent of Adolescent, Petty Trader**

Parents identified facilitators for discussing sexual and reproductive health (SRH) issues with adolescents categorized as adolescent factors, eliminating barriers, and parents' inclusion in sex education. Adolescent factors, such as maturity, closeness, and initiation of discussions, facilitate easier communication. Eliminating barriers and integrating parents into SRH

discussions in school are also suggested. Cultural and religious barriers which impede free sexuality conversations should also be eliminated.

## Discussion

This study in Rivers State, Nigeria, aimed to determine the proportion of adolescents receiving sex education from their parents, explore perceived barriers to sexual-risk communication, and identify factors that may facilitate parent-adolescent sexual-risk communication, highlighting the importance of sex education in adolescents' sexual and reproductive health. Findings revealed a reasonable proportions of adolescents never received sex education and their perceived barriers.

This study reported that 21.0% of the parents had never discussed sex education with their children; and a one-third (32.5%) of the adolescents were more comfortable discussing sexual-related issues with their mothers when communication occurs. A lower proportion was found in a survey done in southern Ethiopia [12] which revealed only a low proportion of adolescents communicated with their parents on SRH issues. In corroboration, a study carried out in Nigeria revealed that the prevalence of communication between parents and children is low on SRH topics such as HIV/AIDS, family planning, and contraception, at 37.4%, 32.5%, and 9% respectively [26]. Similarly, another study conducted in Ebonyi State, Nigeria, also revealed that the majority of adolescents never discussed anything related to sex with their parents or caregivers but discussed it with their friends or peers [27]. The similarities in these studies may be due to the barriers associated with parent-adolescent sexuality communications. This low parent-adolescent sexual and reproductive health communication can make adolescents more prone to diverse negative sexual and reproductive health outcomes.

This study explores the barriers to effective sexual and reproductive health communication among adolescents. Adolescents identified three main factors: parental factors (being too busy, judgmental, poor knowledge on sex education, belief that discussion initiates sex), individual factors (lack of trust, fear/discomfort with discussion), and religious/cultural factors. Both male and female adolescents agreed that their parents were too busy to initiate sexual and reproductive health communications with them. This is consistent with a qualitative review in East Africa and South-Africa, where parents had no time to interact with adolescents due too busy work schedule [28,29]. Adolescents felt disconnected from their parents due to their busy schedules, highlighting the need for work-life balance. Information on sex education was obtained from schools, peers, and internet sources, similar to findings in Ghana and South Africa [29,30].

Female adolescents often avoid discussing sexual matters with their parents due to judgmental attitudes, fearing they are engaging in sex or initiating sexual intercourse, while male adolescents faced harsh feedback on condom use and pregnancy prevention questions. A study in South Africa found that female adolescents found it difficult to confront their parents after discussing sexual matters [29]. These judgmental attitudes led to adolescents relying on other sources of information and never initiating such conversations again. Health education could help correct these issues and promote healthier sexual behavior.

Adolescents often believe their parents have poor or inaccurate knowledge about sex education, leading them to rely on updated sources like the internet and social media. Studies in Ibadan and Ebonyi State, Nigeria, show that parents often feel their children thought they are "old school", may judge them when discussing sexuality issues and don't know what to discuss when it comes to sex-related matters [19,31]. Additionally, parents often lack the right answers to questions on sex, which can hinder sexuality communications [32]. These findings suggest the need for enhanced sex education training for parents and guardians.

Adolescents often trust their parents less with sexual-risk communications, with females often feeling untrustworthy due to their close relationships. Parents fear they're already sexually active or may lead to future experimentation. This aligns with a study revealing barriers to sexual communication, with sex education perceived to lead to early sexual debut [32]. Some adolescents only had sexual and reproductive health communication on physical development and menstrual hygiene. They also felt uncomfortable engaging in sexuality discussions, highlighting the need for health education and improved parent-child relationships to overcome these trust and misconception barriers.

Both genders reported that their parents' cultural and religious beliefs hinder sex education, with adolescents often refusing to engage in discussions due to cultural taboos and religious beliefs. This aligns with a study in Ghana, where rural adolescents often avoid communication about sexual issues with their parents due to these cultural and religious norms [33].

Parents face challenges in discussing sexual and reproductive health with their adolescents, with trader parents disagreeing, while teachers and civil servants agreeing to initiate sexuality education. Civil servants are more likely to initiate sexuality communication as shown in a study [34], possibly due to exposure from enlightened organizations. Lack of communication can lead to more harmful outcomes, and those in agreement believe that a lack of communication could result in more harmful sexual and reproductive outcomes.

Parents face barriers to sexuality communication with adolescents; some of the barriers were shame, gender barriers, and limited education, fear of children experimenting sexually, feeling that children are too young to be educated and tradition/religious beliefs. Parents expressed being shy and having discomfort initiating sexuality and reproductive health discussions In South Africa, most parents feel uncomfortable discussing sexual issues with their children, fearing they will be misconstrued as wanting to engage in sexual activity [29]. Health educations could help bridge these communication barriers, as one female parent never received such discussions from her mother.

Gender barriers were outlined as one of the hindrances parents face to effective discussion on sexual and reproductive health issues with their adolescents. Parents find it difficult to discuss with their children of the opposite gender sexual and reproductive health topics such as condom use, puberty, and physical development. This poses a problem as parents of both sexes tend to communicate with children of the same gender; fathers would rather discuss with their sons, while mothers do the same for their daughters. In corroboration, a qualitative review conducted in East Africa reported gender barriers in that mothers and fathers of adolescents in this region discuss more with adolescents of the same sex because both parents feel shy and hence find it difficult to openly talk to their children [28]. This poses a need to address gender norms and stereotypes as it creates a stumbling block to parent-child sexual-risk communications.

Parents often struggle with discussing sexual and reproductive health issues with their children due to work responsibilities. Those with higher education or corporate experience discuss these issues more than those with lower education. This aligns with studies that found that parents with a tertiary level of education were more likely to initiate sexual and reproductive health communication than those with a primary level of education [33] and have more patience to talk verbally and face-to-face with their adolescents [28]. More health education and enlightenment campaigns could help break these barriers and promote healthier communication between parents and adolescents.

Most parents fear initiating sexuality education with adolescents, believing that discussions about pregnancy, safe sex, and condom use may encourage sexual activities. They believe their children are too young to understand these topics and believe they will learn them naturally as they grow into adulthood. This is similar to studies, where most parents cited

discussing topics like puberty, condom use, and contraceptives as inappropriate due to their children's young age [29,31]. However, effective interaction on sexual and reproductive health can reduce adolescent risk-taking sexual behaviors when parents discuss these issues with adolescents.

Cultural and religious beliefs can hinder effective parent-adolescent sexual risk communication. Some parents avoid discussing sexual and reproductive health matters with their children due to their traditional and religious beliefs. This is similar to studies in South Africa and East Africa, where parents are hesitant to discuss sexuality issues due to traditional norms and cultural beliefs [28,29]. In the same vein, Malaclane and Beckmeyer [34] revealed parental barriers to include: racial/ethnic and religious factors [34]. This may indicate their inability to break free from these norms, potentially leading to sexual exploration. Therefore, changing the culture of silence and educating adolescents on sexual and reproductive health issues is crucial.

Parents suggest several strategies to facilitate easier parent-adolescent sexual and reproductive health discussions. These include prior discussions, church permission, removing barriers, initiation of communication by adolescence, adolescents' maturity, closer relationships, and integration of parents in school discussions. Adolescence is a crucial time for socialization, and parents are the earliest and critical influencers on children's sexual development [35]. They can guide their children's sexual and reproductive health development and encourage safe sexual behavior. To facilitate these discussions, parents must learn to interact socially with adolescents.

## Conclusion

This study found that about a quarter of parents never discussed sex education with their children, while one-third preferred discussing sexual-related issues with their mothers. Over half received sexual education from school, while others received it from home, the internet, or social media. Adolescents perceived barriers to sexual risk communication, including fear, judgment, lack of trust, belief in early sexual initiation, inaccurate knowledge, and cultural or religious barriers. Parents also faced barriers such as gender differences, shame, fear of experimentation, and inadequate knowledge of sexual risk topics. Understanding these barriers could improve parent-adolescent sexuality communications in Rivers State and Nigeria.

This study had its limitations, due to the sensitivity of the topic; some of the adolescents were shy during the FGD. FGDs are often not used for discussing sensitive topics and experiences which is a limitation of the study. The parents that engaged in the in-depth interview gave the same responses; thus, we achieved saturation on time, hence the small sample size. The initial target sample size for this study was calculated to be 394 participants. However, due to incomplete responses in the self-administered questionnaires, the final number of usable responses was reduced to 329. This shortfall represents a limitation of the study, as incomplete responses could not be included in the analysis. Finally, the quantitative study was conducted among in-school adolescents and may not be generalizable for out-of-school adolescents. Despite these limitations, the study contributed a lot to the literature by expounding on parent-adolescent barriers to sexual risk communication in Nigeria.

We recommend that adolescents should be encouraged to openly and honestly discuss sex with their parents to reduce risky behaviour. Parents should be trained to initiate sexuality education, especially at early adolescence. The government should organize training to help parents overcome barriers to discussing sexual health issues and communicate effectively with their children. Addressing socio-cultural norms and religious beliefs can also help. School health programs to train parents and adolescents on overcoming communication barriers are recommended.

## Supporting information

**S1 Text. Transcription of FGD (Males and Females).**
(DOCX)

**S2 Text. Transcription of IDI.** Transcribed data for adolescent participant (15–19 years). Responses to FGD questions raised. Transcription of IDI question responses for parents of adolescents.
(DOCX)

**S1 File. Interview guides.** IDI and FGD questions for adolescents and parents of adolescents respectively.
(DOCX)

**S1 Appendix. Questionnaire.** Questionnaire data collection tool for in-school adolescents aged 15–19 years.
(DOCX)

**S1 Checklist. Code book.** Code book for FGD of in-school adolescents and IDI of parents of adolescents respectively.
(DOCX)

## Author contributions

**Conceptualization:** Chidinma Joycelyn Okpalaku.

**Data curation:** Chidinma Joycelyn Okpalaku.

**Formal analysis:** Chidinma Joycelyn Okpalaku.

**Funding acquisition:** Chidinma Joycelyn Okpalaku.

**Investigation:** Chidinma Joycelyn Okpalaku.

**Methodology:** Chidinma Joycelyn Okpalaku.

**Project administration:** Chidinma Joycelyn Okpalaku.

**Resources:** Chidinma Joycelyn Okpalaku.

**Supervision:** Chigozirim Ogubuike.

**Writing – original draft:** Chidinma Joycelyn Okpalaku.

**Writing – review & editing:** Chigozirim Ogubuike.

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
