## [Decision Letter · Decision Letter 0]

3 Jul 2024

PGPH-D-24-00489

Exploring Barriers to Parent-Adolescent Sexual-Risk Communication among Adolescents in Port Harcourt Nigeria: Adolescents’ and Parents’ Perspective

Dear Dr. Okpalaku,

Thank you for submitting your manuscript to PLOS Global Public Health. After careful consideration, we feel that it has merit but does not fully meet PLOS Global Public Health’s publication criteria as it currently stands. Therefore, we invite you to submit a revised version of the manuscript that addresses the points raised during the review process.

Editor comments:

Please see below for my detailed comments. Please also make sure to address the specific questions and comments made by Reviewers 2 and 3 in the attachments.

We look forward to receiving your revised manuscript.

Kind regards,

Marie A. Brault, PhD

Academic Editor

Journal Requirements:

1. In the online submission form, you indicated that "The data that support the findings of this study are not openly available for ethical reasons (underage human participants), and are available from the corresponding author upon reasonable request". 

3. Uploaded as supplementary information.

Additional Editor Comments (if provided):

- Authors should consider reworking the introduction—there is a lot of repetition that could be reduced. It would also be helpful to situate parent-youth communication and youth SRH in the Nigerian context. What is the situation in Nigeria, and what is the gap in the literature that this study will fill? It also isn’t clear why/how the authors chose this setting for the study.

- I would also encourage the authors to avoid overgeneralizing in their review of the literature in the introduction (for example, “Parental and societal distaste to sex education hasn’t changed over time…”—where is this the case? In certain countries/settings, there is broad support for school-based sex education.)

- In the methods section, it would help to explain why the explanatory sequential design was chosen, as opposed to other approached. In addition, I would encourage the authors to review the COREQ guidelines for reporting qualitative studies, and use this to help in structuring the qualitative component of the manuscript. More information is needed on the qualitative component of the study (how participants were identified and recruited, why IDIs vs. FGDs, qualitative sample size, how thematic analysis was conducted and by whom, etc.), and aligning your revisions with COREQ or a similar qualitative reporting guideline will help. Please include copies of the interview and discussion guides, qualitative codebook(s), and survey instruments as supplemental files.

- Under the ethical consideration section, it would be helpful to clarify why school principles served as guardians for providing consent, instead of parents.

- The results are extremely long. I would encourage the authors to consider re-organizing and streamlining the results. In addition, it would be helpful to provide the readers with an introductory paragraph to the results to orient them. This would also be a good place to note the primary qualitative themes identified.

- The integration of quant and qual data should be clarified in the methods and results. Right now, it isn’t clear whether/how these were integrated in line with the named mixed methods design.

- The qualitative results could use a little more “scaffolding” or explanation. Right now, it reads a little bit like a list of themes and quotes, requiring the reader to make connections.

- The discussion is also very long and could use some reorganizing as suggested by the reviewers.

- In the limitations section, you note that some of the youth were shy discussing the topic, but were assured of confidentiality. However, confidentiality cannot be assured in a FGD, which is why FGDs are often not used for discussing sensitive topics and experiences. I would suggest that the authors include this as a limitation. The limitations of the quantitative data also need to be addressed.

Reviewers' comments:

Reviewer's Responses to Questions

**Comments to the Author**

1. Does this manuscript meet PLOS Global Public Health’s publication criteria ? Is the manuscript technically sound, and do the data support the conclusions? The manuscript must describe methodologically and ethically rigorous research with conclusions that are appropriately drawn based on the data presented.

Reviewer #1: Yes

Reviewer #2: Yes

Reviewer #3: Yes

2. Has the statistical analysis been performed appropriately and rigorously?

Reviewer #1: Yes

Reviewer #2: Yes

Reviewer #3: Yes

3. Have the authors made all data underlying the findings in their manuscript fully available (please refer to the Data Availability Statement at the start of the manuscript PDF file)?

Reviewer #1: No

Reviewer #2: Yes

Reviewer #3: Yes

4. Is the manuscript presented in an intelligible fashion and written in standard English?

Reviewer #1: Yes

Reviewer #2: Yes

Reviewer #3: Yes

5. Review Comments to the Author

Reviewer #1: Congratulate researchers for targeting a very important timely topic and appreciate the effort

However there are some components to be improved

Introduction needs to made in more scientific manner in a order- back group, what is known, gap, how present study fulfilled the gap.

Methods part not clear-As study has different components-for each component study populations and how sampling done should be more elaborated. Class level section not clear for reader.

Discussion-Need to start with brief summary of major findings. Findings are well compared with other research.

Limitations are not discussed adequately

Reviewer #2: The manuscript requires minor revision. mode of questionnaire administration needs to be clear. Provide the ethical approval number/details.

Some grammatical errors would require editing and correction. Also, in the result section, it's important for the author to put the respondents' words in quotation marks so we know we are hearing the voices of the participants

Reviewer #3: I wish to commend the authors seriously for the efforts put in to develop this article. The topic: Exploring Barriers to Parent-Adolescent Sexual-Risk Communication among Adolescents in Port Harcourt Nigeria: Adolescents’ and Parents’ Perspective is an interesting one, and insights have been revealed.

Title: Good

Abstract: Good

Background: The background can be worked on to reduce repetition of the implications of risky sexual behaviour. Most importantly justify the need for the study. How does this study differ from other studies on this topic? What gap in literature do you aim to fill?

Line 40 -42: Sexual curiosity and sexual desire are two different words. Hence, the previous statement those not transcend into the next.

Line 45: Adolescent that are sexually active means that the adolescents are engaging in sexual activity already. Early sexual activity should be struck out from the list of the implications.

Line 48 & 49: The issues should be listed out; using the term several issues is vague.

Line 53: Other non-risky sexual behaviours e.g. engaging in safe sex and abstinence still marriage should be included in the list.

Line 59-60: “Problems associated with” is more appropriate.

Line 78-79: from both the adolescent and parents perspective

Methods: Was permission received from the Ministry of Education/health? This was not stated.

Line 148: Recruitment of participants

Results: Correct grammatical errors in quotes. Edit all “I” to be in capital letter. Once a phrase has been abbreviated, stick to the abbreviations in the later text e.g. SRH. This was also repeated in the discussion section.

Line 257: some from my friends too.

Line 267: that is why I am asking…

Line 271: strike out “health”

Lines 278, 279 & 280 should come at the beginning of the section for parents are too judgemental. Same should be replicated for subsequent sections.

Line 305: the first time

Line 518: may not understand the importance of the discussion

Line 528 & 529: is a repetition. The point has been earlier explained at the beginning of the section

Line 542: “conversation” not “conservation”

Line 562, 563 & 564: and repetition

Discussion: To contextualize the study, more West African and Nigerian studies should be used for Comparism. Also, take note of the dates to use most recent studies.

Line 663: “increase” not “increases”

Conclusion: The introductory part of the conclusion looks more like the result section. Hence, should be rewritten.

References: Well arranged

6. PLOS authors have the option to publish the peer review history of their article (what does this mean? ). If published, this will include your full peer review and any attached files.

**Do you want your identity to be public for this peer review?** For information about this choice, including consent withdrawal, please see our Privacy Policy .

Reviewer #1: No

Reviewer #2: **Yes: ** Vivian Ifeoma Ogbonna

Reviewer #3: **Yes: ** Uduak Ima Andrew-Bassey

While revising your submission, please upload your figure files to the Preflight Analysis and Conversion Engine (PACE) digital diagnostic tool, https://pacev2.apexcovantage.com/ . PACE helps ensure that figures meet PLOS requirements. To use PACE, you must first register as a user. Registration is free. Then, login and navigate to the UPLOAD tab, where you will find detailed instructions on how to use the tool. If you encounter any issues or have any questions when using PACE, please email PLOS at figures@plos.org. Please note that Supporting Information files do not need this step.

---

## [Decision Letter · Decision Letter 1]

2 Dec 2024

PGPH-D-24-00489R1

Exploring Barriers to Parent-Adolescent Sexual-Risk Communication among Adolescents in Port Harcourt Nigeria: Adolescents’ and Parents’ Perspective

Dear Dr. Okpalaku,

Thank you for submitting your manuscript to PLOS Global Public Health. After careful consideration, we feel that it has merit but does not fully meet PLOS Global Public Health’s publication criteria as it currently stands. Therefore, we invite you to submit a revised version of the manuscript that addresses the points raised during the review process.

Editor comments:

We appreciate the authors' responsiveness to previous reviews, and feel the manuscript is much improved.However, the discrepancy between the sample size calculated vs. the actual sample size needs to be addressed as noted by the reviewer's comments below. They offer several suggestions for addressing this, and if not possible, then this should be addressed in the limitations section of the manuscript.

We look forward to receiving your revised manuscript.

Kind regards,

Marie A. Brault, PhD

Academic Editor

Journal Requirements:

Additional Editor Comments (if provided):

Reviewers' comments:

Reviewer's Responses to Questions

**Comments to the Author**

1. If the authors have adequately addressed your comments raised in a previous round of review and you feel that this manuscript is now acceptable for publication, you may indicate that here to bypass the “Comments to the Author” section, enter your conflict of interest statement in the “Confidential to Editor” section, and submit your "Accept" recommendation.

Reviewer #2: (No Response)

2. Does this manuscript meet PLOS Global Public Health’s publication criteria ? Is the manuscript technically sound, and do the data support the conclusions? The manuscript must describe methodologically and ethically rigorous research with conclusions that are appropriately drawn based on the data presented.

Reviewer #2: Yes

3. Has the statistical analysis been performed appropriately and rigorously?

Reviewer #2: Yes

4. Have the authors made all data underlying the findings in their manuscript fully available (please refer to the Data Availability Statement at the start of the manuscript PDF file)?

Reviewer #2: Yes

5. Is the manuscript presented in an intelligible fashion and written in standard English?

Reviewer #2: Yes

6. Review Comments to the Author

Reviewer #2: Your study respondents (329) are below your estimated minimum sample size (394) and should be addressed. The implications are far-reaching.

I advise you to apply a correction formula to your estimated sample size, which should reduce your sample size.

So that the 329 can be valid.

It is important that you address this. I missed pointing this out during the first review. However, it is better to make the correction.

7. PLOS authors have the option to publish the peer review history of their article (what does this mean? ). If published, this will include your full peer review and any attached files.

**Do you want your identity to be public for this peer review?** For information about this choice, including consent withdrawal, please see our Privacy Policy .

Reviewer #2: **Yes: ** Vivian Ifeoma Ogbonna

While revising your submission, please upload your figure files to the Preflight Analysis and Conversion Engine (PACE) digital diagnostic tool, https://pacev2.apexcovantage.com/ . PACE helps ensure that figures meet PLOS requirements. To use PACE, you must first register as a user. Registration is free. Then, login and navigate to the UPLOAD tab, where you will find detailed instructions on how to use the tool. If you encounter any issues or have any questions when using PACE, please email PLOS at figures@plos.org. Please note that Supporting Information files do not need this step.

---

## [Decision Letter · Decision Letter 2]

27 Dec 2024

Exploring Barriers to Parent-Adolescent Sexual-Risk Communication among Adolescents in Port Harcourt Nigeria: Adolescents’ and Parents’ Perspective

PGPH-D-24-00489R2

Dear Miss Okpalaku,

We are pleased to inform you that your manuscript 'Exploring Barriers to Parent-Adolescent Sexual-Risk Communication among Adolescents in Port Harcourt Nigeria: Adolescents’ and Parents’ Perspective' has been provisionally accepted for publication in PLOS Global Public Health.

Best regards,

Marie A. Brault, PhD

Academic Editor

Reviewer Comments (if any, and for reference):

Reviewer's Responses to Questions

**Comments to the Author**

1. If the authors have adequately addressed your comments raised in a previous round of review and you feel that this manuscript is now acceptable for publication, you may indicate that here to bypass the “Comments to the Author” section, enter your conflict of interest statement in the “Confidential to Editor” section, and submit your "Accept" recommendation.

Reviewer #2: All comments have been addressed

2. Does this manuscript meet PLOS Global Public Health’s publication criteria ? Is the manuscript technically sound, and do the data support the conclusions? The manuscript must describe methodologically and ethically rigorous research with conclusions that are appropriately drawn based on the data presented.

Reviewer #2: Yes

3. Has the statistical analysis been performed appropriately and rigorously?

Reviewer #2: Yes

4. Have the authors made all data underlying the findings in their manuscript fully available (please refer to the Data Availability Statement at the start of the manuscript PDF file)?

Reviewer #2: Yes

5. Is the manuscript presented in an intelligible fashion and written in standard English?

Reviewer #2: Yes

6. Review Comments to the Author

Reviewer #2: The authors have addressed my concerns significantly. However, the highlighted section in the manuscript (lines 161 to 166) should be deleted. It is a repitition of what has been stated in the section on ethical clearance.

7. PLOS authors have the option to publish the peer review history of their article (what does this mean? ). If published, this will include your full peer review and any attached files.

**Do you want your identity to be public for this peer review?** For information about this choice, including consent withdrawal, please see our Privacy Policy .

Reviewer #2: **Yes: ** Dr Vivian Ifeoma Ogbonna
